



# Latitudinal variation of Pc3-Pc5 geomagnetic pulsation amplitude across the dip equator in central South America

Graziela Belmira Dias da Silva[1], Antonio Lopes Padilha[1], Lívia Ribeiro Alves[1]

[1]Space Geophysics Division, National Institute for Space Research (INPE), Sao Jose dos Campos, 12227-010, Brazil
*Correspondence to*: Graziela B. D. Silva (graziela.silva@inpe.br; grazybdias@gmail.com)

**Abstract.** In order to clarify the equatorial electrojet effects on ground magnetic pulsations in central South America, we statistically analyzed the amplitude structure of Pc3 and Pc5 pulsations recorded during quiet to moderately disturbed days at multiple equatorial stations nearly aligned along the 10° magnetic meridian. It was observed that Pc3 amplitudes are

attenuated around noon at the dip equator for periods shorter than ~35 s. It is proposed that daytime Pc3s are related to MHD compressional waves incident vertically on the ionosphere, with the screening effect induced by enhanced conductivity in the dip equator causing wave attenuation. Daytime Pc5s showed amplitude enhancement at all equatorial stations, which can be explained by the model of waves excited at higher latitudes and propagating equatorward in an Earth-ionosphere waveguide. However, a slight depression in Pc5 amplitude compared to neighboring equatorial stations and a phase lag in

relation to an off-equatorial station were detected at the dip equator. This result cannot be explained by the ionospheric waveguide model alone and we propose that an alternative propagation model that allows ULF waves to penetrate directly from the magnetosphere to low latitudes could be operating simultaneously to produce these features at the dip equator. Significant effects of the sunrise terminator on Pc3 pulsations were also observed at the stations closest to the dip equator. Contrary to what is reported at other longitudes, in central South America the sunrise effect increases the H-to-D amplitude

ratio. We suggest that these differences may arise from the unique characteristics of this sector, with a strong longitudinal variation in the magnetic declination and precipitation of energetic particles due to the presence of the South Atlantic Magnetic Anomaly. The H-component amplification can be explained by enhancements of the zonal electric field near the magnetic equator driven by F-region neutral winds during sunrise.


**Keywords: Geomagnetic pulsations; Equatorial electrojet; Sunrise terminator.**



# 1 Introduction

Magnetic pulsations, or ULF waves, contain information that can be used to study different physical processes acting in the Sun-Earth system. They can be generated internally to the magnetosphere by various types of disturbances, such as substorms or instabilities associated with cyclotron resonances or drift-bounce resonances, or externally by disturbances in the magnetopause (McPherron, 2005). Its detection on the ground at very low latitudes shows that a significant portion of the energy of these hydromagnetic waves penetrates deep into the magnetosphere and the plasmasphere. Particularly at these latitudes, since ground-based magnetometers respond mainly to changes in E-region currents, pulsations bring records of equatorial ionosphere disturbances associated with complex electrodynamic changes during magnetospheric energy transfer processes.

It is well known that the very low latitude region is characterized by high zonal ionospheric conductivity alongside the dip equator. The most obvious effect of this increased conductivity is that in a narrow latitude range, centered on the dip equator, the eastward flow of dynamo current is intensified during daytime. This enhanced current is known as equatorial electrojet (EEJ). The effect of the EEJ appears as a significant increase of amplitude in the horizontal component of diurnal quiet-time geomagnetic variations, Sq (H), but also in a wide range of short-period fluctuations, which can be observed within a few degrees of latitude on the ground. In the typical pulsation period band, the observed equatorial enhancement close to local noon is ~2-3 times when compared with off-equatorial stations, with a proposed fast decrease for periods smaller than 20 s (Sarma and Sastry, 1995; Shinohara et al., 1997; Zanandrea et al., 2004).

The mechanisms involved in the generation, propagation and amplification of equatorial pulsations have not yet been clearly identified. At very low latitudes, a significant portion of the geomagnetic field lines lies within the ionosphere and therefore the field line resonance theory that explained numerous observations at middle and high latitudes cannot be applied (Yumoto, 1986). Generally, two models are proposed as the possible mechanisms for the generation and propagation of magnetic pulsations near the ground magnetic equator. In the first model, compressional hydromagnetic waves, which propagate from the magnetosheath across the dayside magnetosphere, arrive vertically at the equatorial ionosphere and are interconnected with magnetic disturbances observed on the ground through the ionosphere. It has been proposed that this mechanism is the main source of Pc3 pulsations at low latitudes (Yumoto and Saito, 1983). In the second model, electric fields driven by Alfvén waves in the high latitude ionosphere propagate horizontally through the atmosphere to low latitude regions and the equator in a waveguide bounded by the conducting ionosphere and the ground (Kikuchi and Araki, 1979). Global observations of Pc5 have been interpreted using this equatorward transmission mechanism of the electric field originated in the polar region (Trivedi et al., 1997). The first model would produce an attenuation of pulsation amplitude at the dip equator as a result of shielding effects, while the second model would generate an equatorial enhancement due to the concentration of ionospheric currents at the magnetic equator (Hughes and Southwood, 1976; Shinohara et al., 1997; Tanaka et al., 2004).





Another remarkable feature of the magnetic field in equatorial latitudes is related to the strong longitudinal gradient of ionospheric conductivity near the solar terminator. This feature changes the patterns of the ionospheric currents and consequently of the geomagnetic field reaching the ground. Several authors have studied these effects in regular and irregular geomagnetic pulsations, particularly for the dawn terminator (Saka et al., 1982; Saka and Alperovich, 1993; Tanaka et al. 2007, Imajo et al., 2016). These studies have shown that one of the most prominent effects observed around dawn at very low latitudes is a change in pulsation polarization, characterized by the increase in the D/H amplitude ratio that appears simultaneously with the enhancement of E layer ionization at the local sunrise. Saka and Alperovich (1993) interpreted the sudden increase of the D component amplitude as caused by a meridional current focused along the dawn terminator.

Most of these previous results were based on data from a single station (Saka et al., 1982; Saka and Alperovich, 1993) or only two stations, one located near the dip equator and the other outside that region (Sarma and Sastry, 1995). Consequently, a detailed mapping of the latitudinal dependence of the pulsation amplification and polarization characteristics at very low latitudes has not yet been performed. In this paper, the horizontal spatial structure of equatorial Pc3 and Pc5 pulsations is investigated using data acquired by a meridional magnetometer profile in central South America, with five stations around the dip equator (within ±3° dip latitude) and one station located just outside the equatorial region. In addition, the South American longitude sector presents the unique features of a strong longitudinal variation in the magnetic declination and the presence of the South Atlantic Magnetic Anomaly (SAMA), where energetic particle precipitation is observed due to the global minimum intensity of the geomagnetic field. This variation in the magnetic field strength and declination angle affects the electrodynamic processes of the local equatorial ionosphere (Abdu et al., 2005), which may also be reflected in some of the characteristics of the ground magnetic pulsations in this area.

## 2 Geomagnetic data

The influence of the equatorial ionosphere on the amplitude of Pc3 and Pc5 pulsations is derived here using geomagnetic data recorded at six stations roughly aligned along the 10° magnetic meridian in central South America. A fluxgate magnetometer with self-calibration time system was installed in each of the stations and provided digital data with accuracy better than 0.5 nT in the geomagnetic components H, D, and Z (Tachihara et al., 1996). The signal was recorded with a sampling rate of 3 seconds and an upper limit around 0.7 nT/s was set to ignore fast and very large variations (noise effects or SSC – *Storm Sudden Commencements*). The clock of the data recording system was automatically calibrated by global radio signals (LF OMEGA; Saka et al., 1996), which kept the time accuracy within 100 ms during the data acquisition. The stations operated simultaneously from September to November 1994 and their data were previously used in several studies to investigate different characteristics of geomagnetic variations at very low and equatorial latitudes (e.g., Shinohara et al., 1998; Padilha et al., 2003, 2017; Zanandrea et al., 2004; Rastogi et al., 2008).

Figure 1 shows a map of South America with the location of the six geomagnetic stations. The location of the Huancayo station (to be discussed in the text) is also shown. Relevant information from the geomagnetic field for the period



in which the measurements were performed is presented. These include the dip equator and the inclination of ± 10° that delimit the region where the EEJ currents are expected to affect geomagnetic variations recorded on the ground (Padilha et al., 2003). Important aspects of the geomagnetic field in this region are the large magnetic declination angle and the associated feature that the dip equator crosses the geographic equator and extends deeper down into the South American

5   continent up to ~12°S over the Pacific Ocean coast.

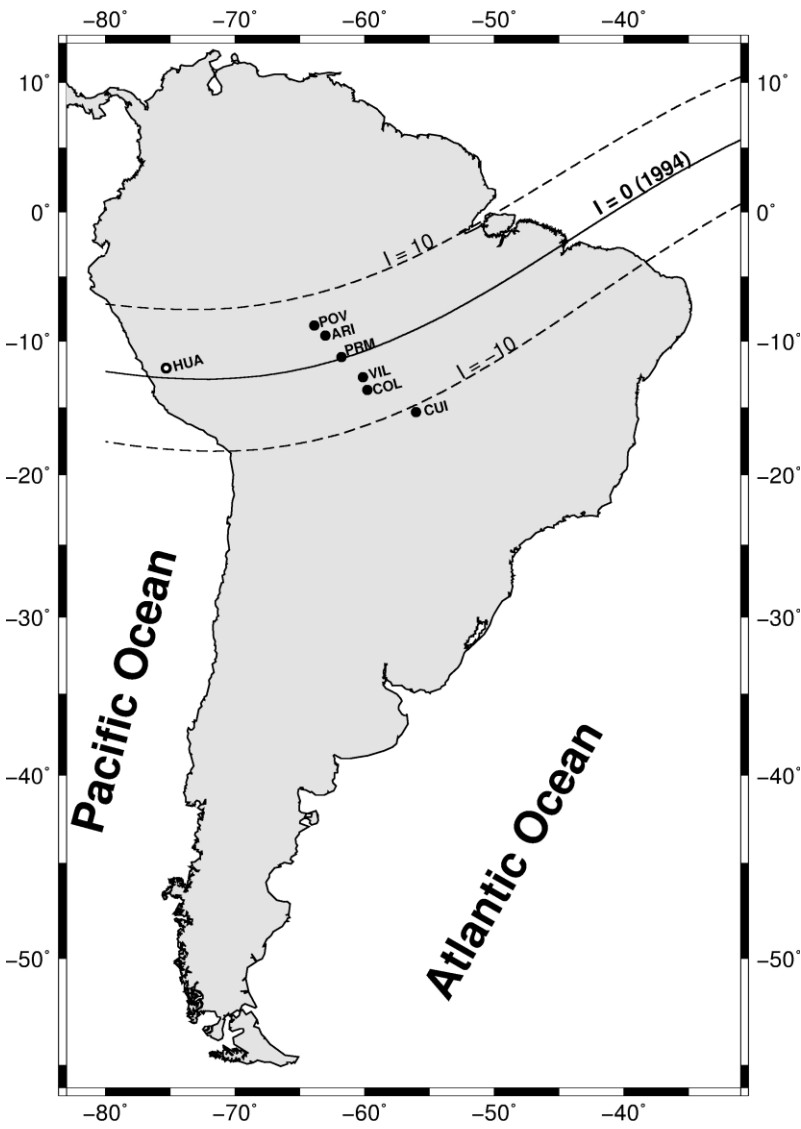

**Figure 1.** Geographic location of the six geomagnetic stations in South America and contours of the geomagnetic field at 1994. The solid
10  line shows the dip equator (I = 0°), and the dashed lines represent ±10° inclination angles. Location of the Huancayo station (HUA) is also indicated. Geomagnetic field values were obtained from the IGRF model.




Five stations were placed in the equatorial region under the influence of the EEJ currents, with one of them at the dip equator and the other four approximately in conjugate points to the north and south of the equator. The latitudinal spacings in the equatorial zone are sufficient to obtain a measurable amplitude change, while maintaining a high level of signal coherence between the stations. The other station (CUI) was located immediately outside the EEJ belt and will be used

here as a reference to evaluate the EEJ effects on the pulsation events.

All stations were positioned well inland, far away from any ocean or mountain range, but a recent study has shown an extensive anomaly in the underground conductivity beneath the CUI site (Padilha et al., 2017). An amplitude enhancement up to tens of percent is observed on the ground geomagnetic variations recorded at this station. Using transfer functions between two nearby geomagnetic stations, inside and outside the anomaly, these authors established the

amplification factors of the geomagnetic field. The amplification varies as a function of frequency due to the reflection of EM waves at the interface with the very good conductor and to differences in damping of the EM wave amplitude during its propagation through the conductive medium inside the Earth. These amplification factors are used here to correct the amplitudes of the geomagnetic variations recorded in CUI.

Geographic and geomagnetic coordinates of the six stations are presented in Table 1. The geomagnetic coordinates

refer to the dipole components of the geomagnetic field, whereas the inclination values are derived from the IGRF. The dip latitude values were calculated from the IGRF inclination values using the Matsushita and Maeda (1965) formula and will be utilized as reference for the geomagnetic station location in the final data analysis. The local noon occurs at nearly 16 UT in all stations.

Table 1 - Geomagnetic and geographic information of the six geomagnetic stations.

| Station | Code | Geographic coordinates | | Geomagnetic coordinates | | L | Magnetic inclination | Dip Latitude |
|---|---|---|---|---|---|---|---|---|
| | | Lat. | Long. | Lat. | Long. | | | |
| Porto Velho | POV | -8.8 | -63.9 | 2.63 | 7.65 | 1.00 | 5.73 | 2.87 |
| Ariquemes | ARI | -9.56 | -63.04 | 1.68 | 8.38 | 1.00 | 3.95 | 1.98 |
| Presidente Médici | PRM | -11.2 | -61.8 | 0.55 | 9.44 | 1.00 | 0.34 | 0.17 |
| Vilhena | VIL | -12.72 | -60.13 | -1.85 | 10.64 | 1.00 | -3.35 | -1.68 |
| Colibri | COL | -13.7 | -59.8 | -2.87 | 10.77 | 1.00 | -5.3 | -2.66 |
| Cuiabá | CUI | -15.35 | -56.05 | -5.64 | 13.89 | 1.01 | -10.76 | -5.43 |

Simultaneous data from the six geomagnetic stations were obtained during 60 days from September 3 to November 1, 1994. Since we are interested in the ionospheric contribution to Pc3 and Pc5 amplitudes, we preferentially considered the data corresponding to geomagnetically quiet or moderately disturbed conditions. Typical solar daily variations (Sq) in the H

component of the geomagnetic fields measured simultaneously by the six stations under quiet solar condition have already been presented by Padilha et al. (2017). It was observed that the diurnal variation has a maximum around noon (~ 16 UT),




with the largest amplitude at the station closest to the dip equator (PRM) and progressively decreasing to the north and south. The stations at conjugate points (pairs ARI-VIL and POV-COL) have a nearly identical diurnal variation profile.

Figure 2 shows the behavior of the geomagnetic field during the measurement period through geomagnetic indices and the activity level of filtered Pc3 and Pc5 pulsations. The Dst index indicates geomagnetically active conditions during
this period with the occurrence of one intense (minimum Dst of -123 nT on October 29) and three moderate magnetic storms (Dst minima between -50 and -100 nT on September 25, October 3 and October 23). The AE index provides a measure of auroral electrojet activity and indicates that, in addition to the disturbed periods during the four magnetic storms, another enhanced substorm activity took place between September 6 and 14. The variance levels of the H-component filtered data in Pc3 and Pc5 frequency bands of the CUI station are also shown to indicate the level of wave activity. It can be seen that the
pulsation activity is concentrated mainly during the disturbed intervals indicated by the AE index. To minimize the contribution of the non-equatorial ionosphere to the pulsation amplitude we discarded the events during the main phase of the magnetic storms. The temporal location of the chosen events is shown in the AE index graph, which indicates that they are mainly concentrated during the substorm activity and in the recovery phases of the magnetic storms.

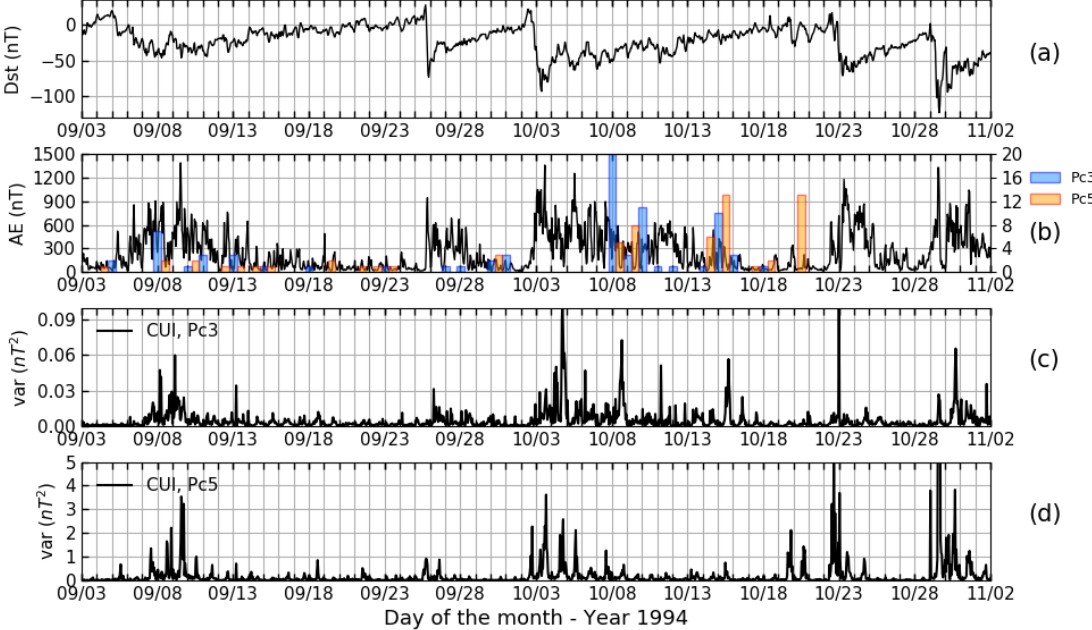

**Figure 2.** Geomagnetic indices and filtered ground geomagnetic pulsations during the period September 3 to November 1, 1994. (a) Dst index; (b) AE index with the temporal location and number of events chosen for the pulsation analysis; (c) variance of Pc3 pulsations in the H-component of the CUI station; (d) variance of Pc5 pulsations in the H-component of the CUI station.





# 3 Data processing

## 3.1 Selection of Pc3 and Pc5 events

The time series data were digitally filtered in Pc3 and Pc5 frequency bands using a recursive Butterworth-band-pass type, with unit gain within the chosen frequency band (Kanasewich, 1981). Since ground pulsations near the geomagnetic

equator are known to be strongly polarized to the H-component, we initially searched for simultaneous pulsations events in this component for time intervals representing daytime (14 – 18 UT corresponding to 10 – 14 LT) and nighttime (22 – 10 UT corresponding to 18 – 6 LT). Daytime data allow us to evaluate the influence of the EEJ on the signals and the time interval around noon is when these currents are more intense (Le Mouel et al., 2006), while nighttime data are used as a reference of signals less affected by the equatorial ionosphere. The hourly variance of the H-component in CUI was used to find the

relevant events, which were later investigated in the equatorial stations. Figures 3 and 4 show examples of Pc3 and Pc5 events simultaneously observed in all stations.

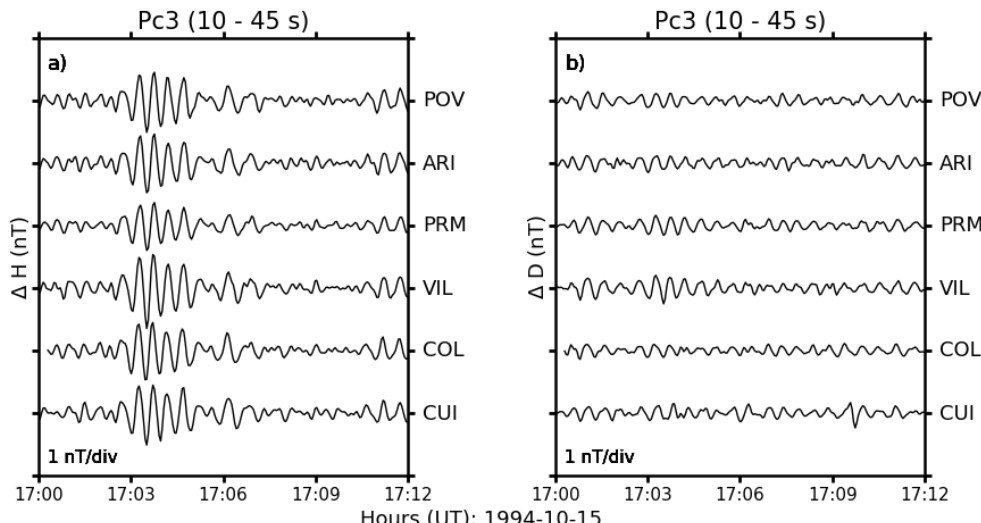

**Figure 3.** Band-pass-filtered geomagnetic variations (H and D components) in the Pc3 interval, shortly after noon on October 15, 1994. A Pc3 event starting at ~1703 UT is observed.

Visual inspection indicates that the Pc3 pulsation is apparently damped at the PRM station (under the dip equator) compared to the other equatorial stations. On the other hand, the Pc5 pulsation is strongly amplified in all equatorial stations in relation to CUI. In addition, the strong equatorial polarization to the H-component is also observed, especially for the Pc5 event. These results agree with those reported previously (e.g., Sarma and Sastry, 1995; Trivedi et al., 1997) and show the

strong effect of EEJ currents on pulsation amplitude around local noon.



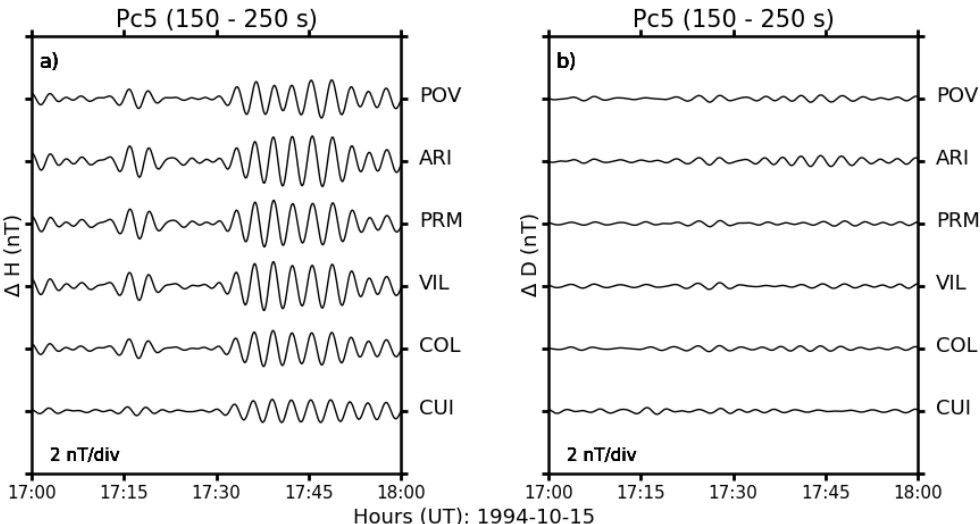

**Figure 4.** The same as Figure 3, for Pc5 pulsations in the afternoon on October 15, 1994. An event is observed in the H-component after 1730 UT.

To emphasize the effect of equatorial ionospheric currents on the pulsation amplitude, we chose days with little magnetic activity in the equatorial region. Only those days where the minimum Dst value was greater than -50 nT were chosen, thus eliminating the main phase of the four magnetic storms. Under these conditions, 30 days with significant signals of Pc3 and Pc5 pulsations were found. In these days, 77 Pc3 and 65 Pc5 clear events were identified.

10 **3.2 Relative amplitude of equatorial Pc3 and Pc5 pulsations**

Following the procedure described in Roy and Rao (1998), the characteristic period of each event was determined from spectral analysis. The CUI station signals were corrected using the period-dependent amplification factors defined by Padilha et al. (2017). The correction factor in the Pc3 frequency band is given by a second-degree polynomial function, whereas a fourth-degree polynomial function was used for the Pc5 band (pulsations in the CUI station shown in Figures 3 15 and 4 were previously corrected for the underground amplification).

The EEJ effect for each event was estimated by the ratio between the spectral amplitude (in units of $nT/\sqrt{Hz}$) at the off-equatorial station (CUI) and the spectral amplitudes of the equatorial stations. Thus, ratios (or relative amplitudes) greater than one correspond to equatorial damping, whereas ratios smaller than one indicate equatorial amplification. Figure 5 exemplifies the procedure for the Pc3 event shown in Figure 3. The left panels (a-f) show the dynamic spectrograms of the 20 Pc3 event for each geomagnetic station, where 0 dB corresponds to 1 $nT^2$/Hz. These power spectral density (PSD) distributions have the same trend identified visually in the time series of Figure 3, with the PRM signal substantially





attenuated in comparison with the other stations. It can also be noted that the VIL station has greater amplitude for this event while the other stations have similar PSD values. The average spectral amplitude for this event at each station is shown on the right (panel g), where the period of peak amplitude in CUI is identified. This period is used as reference to obtain the spectral amplitude of all stations. The ratios between the amplitude in CUI and the other stations are then calculated to derive

5    an amplification (or damping) factor for each equatorial station for that specific event.

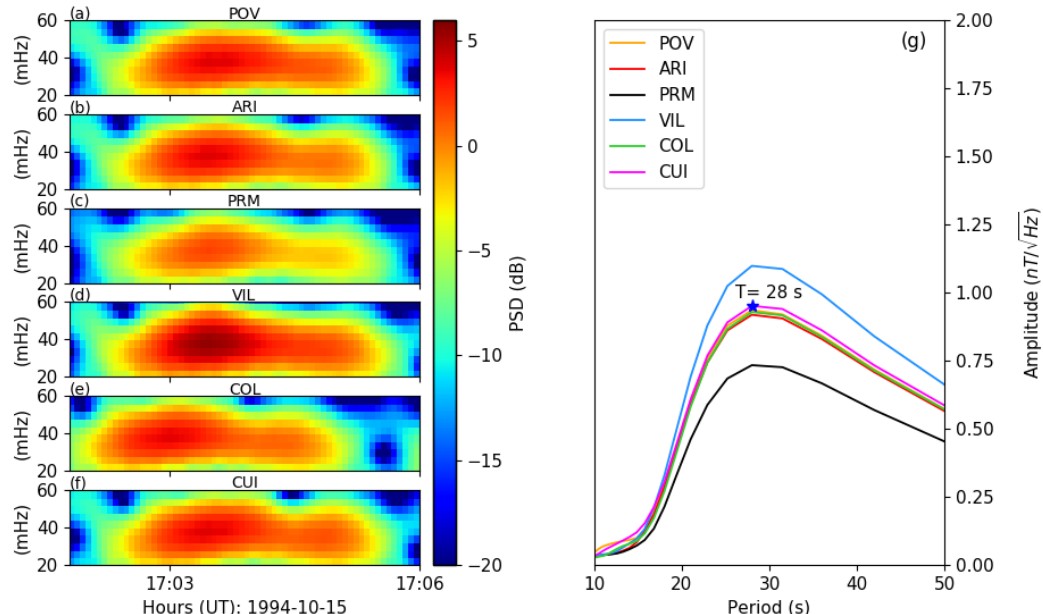

**Figure 5.** Power spectral analysis for the Pc3 pulsation on October 15, 1994 shown in Figure 3. Left panels (a-f) show dynamic spectrograms of power spectral density during the event. Right panel (g) shows the average spectral amplitude at each station with identification of the period of peak amplitude in CUI station.

Figure 6 shows the occurrence distribution of Pc3 and Pc5 relative amplitudes for all events at each equatorial station as a function of local time (LT) and universal time (UT). It can be seen that most of the Pc3 events are slightly amplified in the equatorial region, during both daytime and nighttime. However, there is a larger number of damped events around noon. This is particularly salient for the dip equator (PRM station) where most of the events are significantly

15   damped. An increase in the Pc3 amplification is observed in stations POV, ARI, PRM and VIL shortly before 6 LT. This amplification is associated with the sunrise effect and will be discussed later in greater detail. On the other hand, almost all Pc5 events are amplified at equatorial stations. The highest amplifications are observed during daytime (6 LT to 18 LT) in all stations.



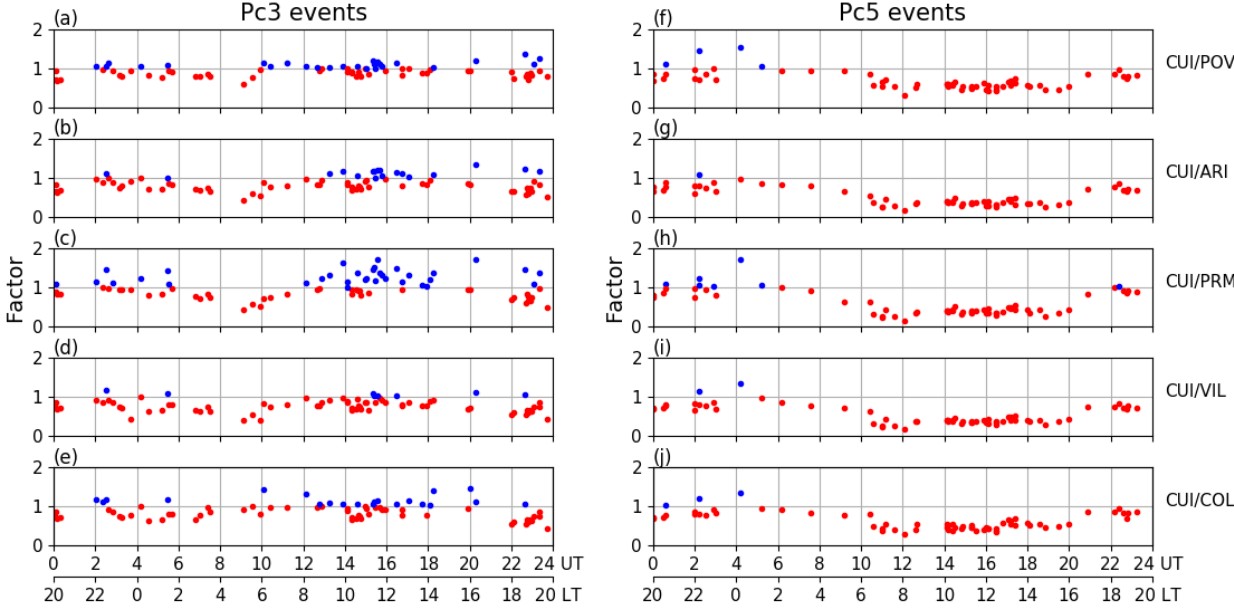

**Figure 6.** Local time dependence of occurrence distributions of Pc3 and Pc5 relative amplitudes at the equatorial stations. In blue are values greater than one (damping), in red are values less than one (amplification).

The relative amplitudes of the Pc3 pulsations were also analyzed as a function of the wave period. For this analysis we used only nighttime events (18 LT to 6 LT) and noon events (10 LT to 14 LT). It can be observed in Figure 7 that nighttime events are not dependent on the period, being preferably amplified throughout the Pc3 band. On the other hand, the noon events present significant differences as a function of the period. It should be noted that, in the latter case, no pulsations with a period shorter than 27 s were observed. Besides, there is an evident cutoff period around 35 s for amplification or damping under the dip equator. Events with periods shorter than 35 s are damped, while longer periods are amplified. This observation explains why some events around noon in Figure 6 show amplification at the PRM station (these events have periods longer than 35 s). In summary, these results show that at the dip equator there is an amplitude damping in Pc3 pulsations with periods of less than 35 s, when EEJ currents are well developed around local noon.



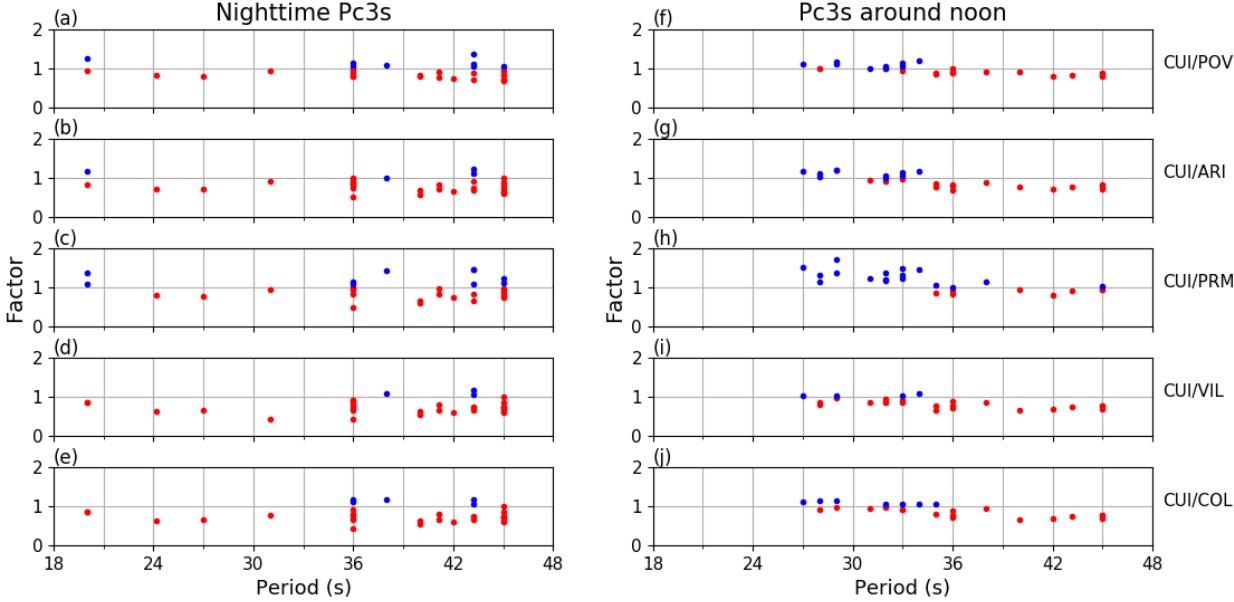

**Figure 7.** Period dependence of the relative amplitudes of nighttime and noon Pc3 pulsations in equatorial stations.

## 4 EEJ effects on pulsation amplitude

We found that Pc3 amplitudes undergo a depression under the dip equator around noon and this is a period-
dependent phenomenon. On the other hand, Pc5 amplitudes are enhanced at all equatorial latitudes, regardless of the period.
During nighttime, Pc3 and Pc5 amplitudes are slightly increased for all stations in the equatorial region. These results from
the central South American region agree with many previous studies at similar latitudes, based on both ground observations
and theoretical models of excitation and propagation of MHD waves that lead to pulsation activity on the Earth's equatorial
surface (e.g., Yumoto et al., 1985; Itonaga and Kitamura, 1993; Roy and Rao, 1998; Takla et al., 2011). However, they
contradict other studies that showed daytime Pc3 amplification in the equatorial region, without specifying any damping
interval (Matsuoka et al., 1997; Zanandrea et al., 2004).

Our results can be explained by considering the mechanisms for generation and propagation of the magnetic
pulsations until their detection on the ground at the equatorial region. As discussed by Yumoto (1986), two models have
been proposed to explain Pc3 waves observed at very low latitudes. In the first model, upstream waves generated by ion-
cyclotron instabilities on the bow shock propagate in the form of compressional waves along the equatorial plane of the
magnetosphere, cross the lines of the magnetic field and arrive directly at the equatorial ionosphere. In the second model,
surface waves generated by instabilities at the diurnal magnetosphere boundary (such as Kelvin-Helmholtz instabilities)
propagate to the high latitude ionosphere and generate large-scale ionospheric current oscillations at these latitudes. These





high-latitude currents leak into low latitudes and can cause Pc3 pulsations near the magnetic equator. Following Yumoto (1986), the latter model cannot explain the occurrence of equatorial Pc3 because they can only be transmitted to the high latitude ionosphere due to the high damping of these waves in their propagation in the radial direction. Therefore, equatorial Pc3 pulsations are more likely to be related to the direct transmission of the compressional fast mode towards the ground.

These waves are usually controlled by the solar wind parameters and the interplanetary magnetic field (IMF), with their transmission into the Earth's magnetosphere connected to small cone angle values (Russell et al., 1983).

Theoretical considerations about variations on pulsation amplitude due to the direct incidence of a plane compressional MHD wave on the equatorial ionosphere were obtained in different studies (e.g., Itonaga and Kitamura, 1993; Itonaga et al., 1998). These models predict a depression in the amplitude of the pulsation on the ground at the dip equator,

since the induced ionospheric currents act as an obstacle in the propagation of compression waves. Such depression arises from the ionospheric screening effect which becomes more marked as the ionospheric conductivity increases. Consequently, these models predict a greater damping in the pulsation amplitude at the dip equator around noon (when the ionospheric conductivity reaches its maximum), as observed in our data. On the other hand, although some of these models take into account the frequency dependence for this pulsation amplitude depression under the dip equator, there is still no clear

physical explanation for the occurrence of a maximum period that limits this damping effect.

Pc5 waves are a persistent component of a disturbed magnetosphere and their possible sources include Kelvin-Helmholtz oscillations in the magnetopause, excitation of cavity or waveguide modes, direct control by oscillations in the solar wind, fluctuation of field-aligned currents in the auroral zone and drift-bounce resonance with ring current particles (Hughes, 1994; Elkington, 2006; Kessel, 2008). They are often observed near auroral latitudes, but can also be observed at

mid- and low-latitudes (Ziesolleck and Chamalaun, 1993). The occurrence of these waves at equatorial latitudes (Reddy et al., 1994; Trivedi et al., 1997) led to the proposition of different penetration mechanisms. Kikuchi and Araki (1979) argued that the ULF wave energy arrives from the magnetosphere to high latitudes first and then propagates to low latitudes through an ionosphere waveguide, while Chi et al. (2001) proposed a combination of fast and shear Alfvén waves to allow ULF waves to penetrate directly from the magnetosphere to low latitudes.

The Earth-ionosphere waveguide model of Kikuchi and Araki (1979) was initially developed to elucidate the simultaneous observation of preliminary inverse impulse (PRI) of the magnetic sudden commencement (SC) at high latitudes and at the magnetic dip equator. It was later expanded to explain the equatorial enhancement of a class of short-period fluctuations, including Pc5 pulsations. In this model, surface waves generated by instabilities at the dayside high-latitude magnetosphere boundary generate large-scale ionospheric current oscillations at the polar ionosphere. These Pc5-related

polar electric fields can propagate to the low-latitude ionosphere at the speed of light through the zeroth-order transverse magnetic (TM0) Earth-ionosphere waveguide mode and can generate zonal ionospheric currents at low latitudes. A numerical simulation by Tsunomura and Araki (1984) has shown that the amplitude of the polar electric field decreases as it propagates towards the equator due to the small proportion of the size of the polar electric field in relation to the propagation distance. Nevertheless, the process is still efficient enough to allow the polar waves to reach the dip equator and be abruptly



amplified by the high ionospheric Cowling conductivity in that region. Thus, this model predicts an enhancement of Pc5 amplitudes in the vicinity of the dip equator.

Using a global network of magnetometers with high time accuracy, Chi et al. (2001) observed differences in the arrival time of PRI signals at middle and low latitudes. This result raised questions about the validity of the Earth–ionosphere waveguide propagation model of Kikuchi and Araki (1979), which predicts that the PRI onset should be seen simultaneously at all locations on the Earth's surface. Chi et al. (2001) suggested that these differences could be explained by MHD wave propagation along the path proposed by Tamao (1964), in which a compressional wave propagates along the equator until it is converted into a shear Alfvén wave that then propagates along the field lines to the ionosphere. This penetration mechanism has also been recently proposed for different ULF waves (e.g., Yizengaw et al., 2013). Due to the screening effect of the enhanced ionospheric conductivity at the equator on an MHD wave signal incident from the magnetosphere, this model would lead to a damping of the wave amplitude at the dip equator and a phase lag of this signal compared to an off-equatorial region.

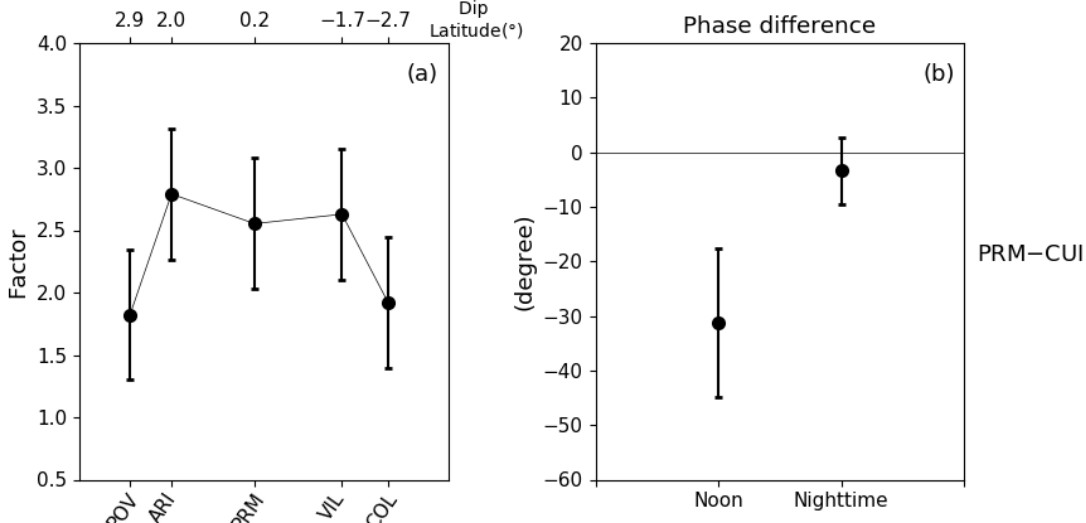

**Figure 8.** (a) Latitudinal profile of amplification factors for Pc5 pulsations around noon. (b) Phase delays between the dip equator station (PRM) and off-equatorial station (CUI) for Pc5 events during nighttime and around noon. Mean and standard deviations are indicated by dots and bars, respectively.

Both characteristics were tested in our dataset. Figure 8a shows the mean relative amplitudes of Pc5 pulsations around noon for the 5 stations in the equatorial region, with standard deviation error depicted by error bars. To facilitate visualization, we are showing the amplitude ratio of the equatorial stations by the reference station (CUI), contrary to that was used in Figures 6 and 7. Although the one-sigma error bars overlap in the different stations, the three stations closest to the dip equator are more amplified, with the lowest amplification observed at the station closest to the dip equator (PRM).





This is a statistically robust result, since it represents a tendency observed in 85% of the events. On the other hand, Figure 8b shows the phase delays between the dip equator station (PRM) and the off-equatorial station (CUI) for Pc5 pulsations recorded during nighttime and around noon. Phase differences for daytime Pc5 have a scattered distribution, ranging from ~ -45° to -20° and indicating a significant phase delay of the equatorial station relative to the off-equatorial station. This result

is in agreement with previous studies in the Brazilian equatorial region (e.g., Shinohara et al., 1998). The nighttime events show less significant phase differences, on average very close to zero. This result is consistent with the absence of significant effects generated by the nighttime equatorial ionosphere. The same model of vertically incident MHD waves used to explain the damping of Pc3 pulsations by the enhanced ionospheric conductivity around noon in the dip equator can also be used to explain the lower amplitude amplification and maximum phase delay observed for Pc5 pulsations.

Generally, the detailed characteristics of the Pc5 amplification and phase delay in the South American equatorial region are not consistent with only one of the proposed wave transmission mechanisms. The Earth-ionosphere waveguide propagation model agrees well with the overall equatorial enhancement of these pulsations, but is inconsistent with the lower amplification observed at the dip equator and with the phase lags reported in the study. On the other hand, the MHD wave propagation model explains the local damping in Pc5 amplitudes and the expressive phase lag at the dip equator, but does

not account for the general equatorial enhancement of the Pc5 signals. Probably, when the two mechanisms work in an interactive coupled manner, they may explain the variability of the equatorial enhancement. A more advanced theoretical model is needed, which would consider the simultaneous effects from MHD wave propagation and from the Earth–ionosphere waveguide. It must be also considered that fine-scale amplification factors in the equatorial zone should be taken cautiously since ground-based magnetometers have poor spatial resolution because the pulsation signal arises from the effect

of ionospheric currents integrated over a transverse ionospheric region of dimensions comparable with the height of the ionosphere (Engebretson et al., 1995).

## 5 The sunrise effect

The local time distribution of Pc3 relative amplitudes in Figure 6 shows an expressive amplification around sunrise at the stations closest to the dip equator (ARI, PRM, and VIL). As an example of such events, Figure 9 shows band-pass

filtered data (10 to 45 s) of geomagnetic H and D components during the interval from 1100 to 1112 UT (0700 to 0712 LT) on October 15, 1994. At variance with the Pc3 events around noon (see Figure 3), this event presents an apparent amplification in the H-component at these three stations when compared to the CUI reference station. The pulsations in D-component at all stations are very small, so that the almost linearly polarized ellipses in the equatorial belt are predominantly oriented in the north-south direction. Consequently, on contrary to the previously reported studies at other longitudes, in the

central South American equatorial region our data show that the sunrise effect on Pc3 pulsations increases the H-to-D (north-south to east-west) amplitude ratio.



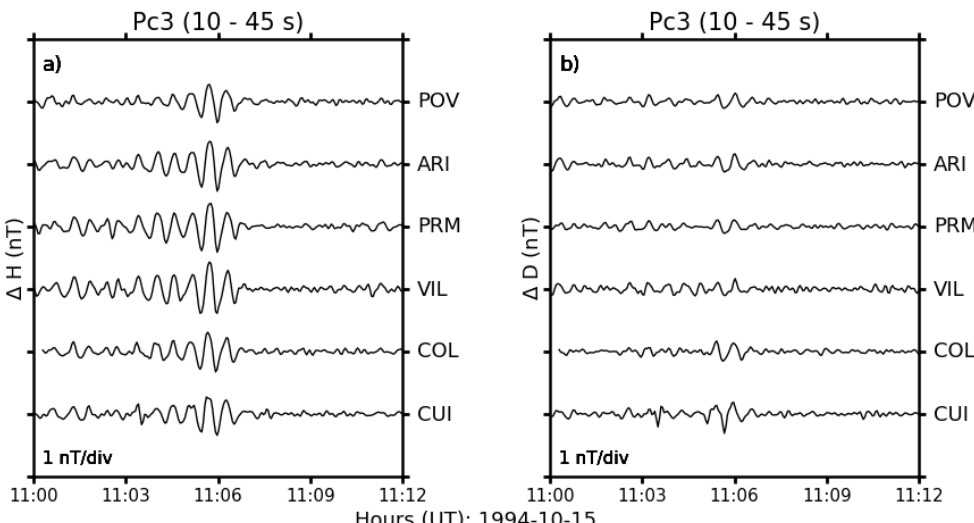

**Figure 9.** Band-pass-filtered geomagnetic variations (H and D components) in the Pc3 interval, shortly after sunrise on October 15, 1994.

Figure 6 presents a small number of events around sunrise. To clarify the significance and effect of the sunrise terminator on Pc3 pulsations, the data were reevaluated to obtain statistically more robust results. Only two stations were used to detect coherent events, one under the dip equator and affected by the dawn effect (PRM) and the other unaffected by this effect (CUI). Using these stations, 32 events were detected between 2 and 10 LT and their relative amplitudes (PRM/CUI) are shown in Figure 10. It can be seen that most of the events were amplified at the station closest to the dip equator, with the largest amplification occurring around sunrise (~ 5-6 LT).



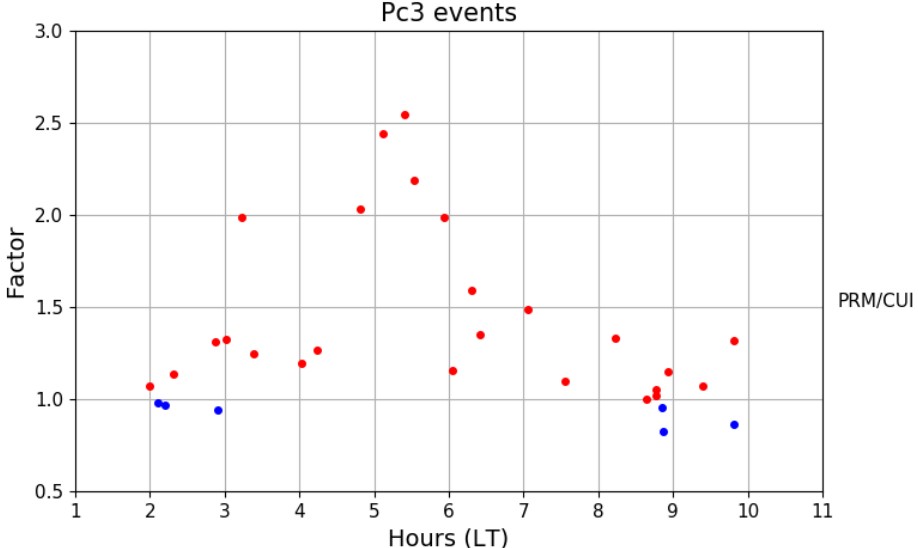

**Figure 10.** Distribution of Pc3 relative amplitudes (PRM/CUI) in H-component between 2 and 10 LT. In red are values greater than one (amplification) and in blue are values less than one (damping).

5   The amplification characteristics observed by our data are certainly related to the strong longitudinal gradient of the ionospheric conductivity near the dawn terminator. This feature changes the patterns of the ionospheric currents and consequently of the magnetic field on the ground driven by these currents. Considering the strong North-South polarization of the ground pulsations, we suggest that an increase of zonal (East-West) ionospheric currents in the sunlit region adjacent to the dawn terminator controls the observed amplification of the H-component. This interpretation is exactly opposite to that

10  commonly proposed as sunrise effect in other studies, which suggest an increase in the meridional component of the ionospheric currents around the dawn terminator. In particular, this result contradicts that of Saka and Alperovich (1993) at the Huancayo station on the South American west coast, very close to the region of our studies (see location in Figure 1). However, in agreement with our results, a number of studies based mainly on satellite observations have reported anomalous enhancements of eastward electric fields near sunrise in the equatorial ionosphere, seasonally correlated with the sign of the

15  magnetic declination at the point of observation (Aggson et al., 1995; Kelley et al., 2014; Zhang et al., 2015).

  An explanation for this difference can be sought in terms of the existence of large longitudinal variations in electrodynamic processes in the South American equatorial ionosphere, as verified by various types of ground- and satellite-based observations. These variations were reviewed by Abdu et al. (2005) and are associated with the strong longitudinal variation in the geomagnetic declination angle and total field intensity due to the presence of the SAMA. The magnetic

20  declination angle controls the development of the F-layer dynamo in dawn and dusk hours that couples the E and F regions and plays a key role in the electrodynamics of the equatorial ionosphere. The enhanced ionization by energetic particle



precipitation increases the background conductivity distribution in the ionosphere over the SAMA central region and may also extend for several degrees in longitude and latitude reaching the equatorial ionosphere, even in quiet conditions. The combination of both effects is a unique feature of the South American region and causes significant longitudinal/seasonal differences in the equatorial plasma density distribution.

The dissimilarities in the results between central South America (PRM station) and West South America (HUA station) may therefore arise mainly from differences in the geomagnetic parameters of the two stations. During the experiment of Saka and Alperovich (1993), the magnetic declination angle and the total field intensity in HUA were $1.5^{o}$E and 27,000 nT, respectively, whereas during our measurements these values in PRM were $10.3^{o}$W and 25,200 nT. This shows that the PRM station is much closer to the central region of SAMA. In this region, azimuthally drifting energetic

particles trapped in the Earth's Van Allen internal radiation belt approach the Earth's surface due to the low values of the geomagnetic field intensity, interacting with the dense atmosphere and producing enhanced ionization at ionospheric E layer heights (see, for example, Paulikas, 1975, for a review).

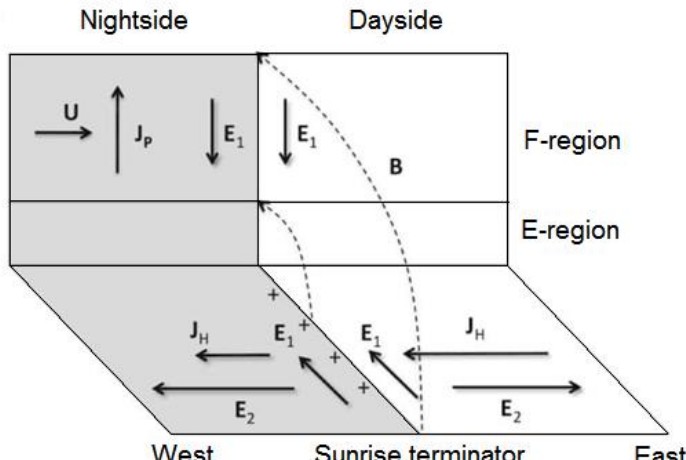

**Figure 11.** Sketch illustrating the processes responsible for the enhancement of zonal electric fields around sunrise (partially modified from Farley et al., 1986; Kelley et al., 2014). Neutral thermospheric winds ($U$) moving eastward in the F-region across the dawn terminator give rise to Pedersen currents ($J_P$) in the F-region, which create the polarization field ($E_1$) that is mapped to the E-region through the magnetic field lines ($B$). Hall currents ($J_H$) through the terminator create an accumulation of positive charges along the terminator and a secondary electric field ($E_2$), directed to the east on the dayside and to the west on the nightside.

      Following Abdu et al. (2005), the extra ionization at PRM due to SAMA develops polarized electric fields in the vicinity of the enhanced conductivity gradient of the dawn terminator, particularly when external electromagnetic fields (e.g., geomagnetic pulsations) are imposed. A secondary longitudinal electric field structure is produced which is superimposed, in phase, on the regular zonal electric field. This enhancement of the zonal electric field around dawn can be



explained following the same mechanism proposed by Farley et al. (1986) for the development of pre-reversal electric field enhancement in the evening hours. In this case, F region thermospheric winds blowing eastward through the sunrise terminator are the source of sunrise enhancement. Figure 11 illustrates the creation of the polarization electric field in the equatorial F region and its mapping to the low-latitude E region (as presented by Kelley et al., 2014). The zonal gradient of

the Hall conductance generates an accumulation of positive charges across the sunrise terminator and, as a consequence, an enhancement of zonal electric fields is developed to the east (dayside) of the terminator.

This enhancement of the integrated zonal electric field attains its largest values when the sunrise terminator is aligned with the magnetic meridian. In central South America, this condition prevails during a period close to the December solstice, when there is a seasonal maximum in the intensity of the secondary zonal electric fields associated with the negative

magnetic declination (Zhang et al., 2015). This is exactly the period in which our measurements were taken. On the other hand, the proposed mechanism for the zonal electric field enhancement over PRM around sunrise would be seasonally dependent on the solar zenith angle. Due to the limited temporal coverage of our ground measurements, we cannot test this hypothesis at PRM, but Saka and Alperovich (1993) reported a seasonal modulation in the polarization parameters at HUA as a function of the solar angle. Also, despite the many evidences of the longitudinal ionospheric variability in the South

American equatorial sector, it is difficult to verify our proposition to explain the observed differences between PRM and HUA due to the lack of direct and simultaneous measurements of ionospheric electric fields at these places.

## 6 Summary and conclusions

Using multipoint observations at equatorial ground stations in central South America, we studied the spatial variation in the amplitude of continuous pulsations (Pc3 and Pc5) observed in the zone of influence of the EEJ currents.

Nighttime events in the equatorial region exhibit a systematic pattern of minor amplification, while pulsation amplitude changes during daytime as a function of the period and as the stations approach the dip equator. These results can be explained by the low intensity of the equatorial ionospheric currents at night and by the development of a high ionospheric conductivity along the dip equator during daytime.

We found attenuation in the Pc3 amplitudes around local noon. This amplitude damping is observed exclusively at

the station closest to the dip equator and for wave periods shorter than ~ 35 s. According to previous results and proposed models of excitation and propagation, the main source of equatorial Pc3 must be related to compressional upstream waves, with the amplitude attenuation associated with ionospheric shielding effects (Yumoto et al., 1985). However, the physical concept behind the occurrence of a maximum period limiting the damping effect remains to be elucidated. On the other hand, Pc5 amplitudes show an enhancement at all equatorial stations, which can be explained by the model of Pc5 waves

excited at higher latitudes and propagating equatorward in a Earth-ionosphere waveguide (Kikuchi and Araki, 1979). However, the availability of several stations operating simultaneously under the EEJ currents allowed to detect a slight depression in the Pc5 amplitude at the dip equator when compared to neighboring stations (within ± 2° of dip latitude). Also,





a phase lag between the Pc5 signals observed at the dip equator and just outside the equatorial region was observed. Both features are not predicted by the horizontal transmission model of Kikuchi and Araki (1979). We propose that the alternative model of MHD compressional waves vertically incident on the ionosphere (Chi et al., 2001) should work simultaneously with the Earth-ionosphere waveguide model to produce Pc5 depression and phase lag at the dip equator.

Another important result is the sunrise effect observed on Pc3 ground pulsations at the stations closest to the dip equator. Its main effect on our data is to increase the H-component, rather than the D-component as reported by studies at other longitudes. We explain this amplification by the enhancement of the zonal electric field around dawn and sought the differences with the other studies in the unique characteristics of our study area, with a strong longitudinal variation in the magnetic declination and occurrence of energetic particle precipitation due to the presence of the South Atlantic Magnetic

Anomaly. It is proposed a mechanism to give rise to the enhanced zonal electric field, based on F region thermospheric winds blowing eastward through the sunrise terminator (Farley et al., 1986; Kelley et al., 2014).

In summary, the study presented new information about the effects of EEJ currents on the geomagnetic pulsations recorded on the ground at equatorial latitudes. The detailed characterization of the Pc5 pulsation amplification in the region around the dip equator was not previously known and was only possible by the availability of several stations operating

simultaneously under the EEJ. Also noteworthy are the results on the peculiar feature of the sunrise effect in this region. A very unique combination of factors contributed to the anomalous behavior of Pc3 pulsations around the dawn terminator. On the other hand, interpretation is limited by the small amount of data with the stations operating simultaneously and by the absence of simultaneous measurements of other ionospheric parameters during the study period to support the proposed interpretation.

**Acknowledgements**

The study was supported by a research grant from FAPESP (92/04764-7) and fellowships from CNPq (304353/2013-2 and 131675/2015-0). The geomagnetic experiment was designed and carried out by T. Kitamura (Kyushu University, Japan) and N.B. Trivedi (INPE).

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
