# Peer review of "Latitudinal variation of Pc3-Pc5 geomagnetic pulsation amplitude across the dip equator in central South America"

_Annales Geophysicae, 2019_

## Referee Comment (RC1) · Anonymous Referee #1 · 31 Jul 2019

This manuscript investigates a latitudinal structure of amplitude and phase of Pc 3 and Pc 5 pulsations observed around the magnetic equator in central South America. It was demonstrated that Pc 3 pulsations with periods shorter than ~35 s are attenuated around local noon at the dip equator and the authors proposed that the amplitude attenuation of dayside Pc 3s is caused by the ionospheric screening effect on MHD compressional waves incident on the equatorial ionosphere. On the other hand, dayside Pc 5 pulsations showed the amplitude enhancement around the equator, with a slight depression at the dip equator compared with those at the neighboring equatorial stations. They proposed a coupled model of the Earth-ionosphere waveguide propagation and the compressional wave propagation. The attenuation of dayside Pc 3s

at the dip equator and the enhancement of dayside Pc 5s around the equator have already been reported by several previous papers, but the detailed spatial structure of the pulsations within 3 degrees dip altitude is new and clear. In addition, this manuscript includes some new findings, i.e., the attenuation in Pc 5 amplitude at the dip equator and the increase in the H-to-D amplitude ratio of Pc 3s at the dawn terminator. Thus, I believe that this paper is suitable for publication in Annales Geophysicae. However, the manuscript is not acceptable in the present form, because there are some points to be improved in the discussion.

Specific comments:

Section 5: To explain the equatorial enhancement of Pc 3 pulsations in the H component at the dawn terminator, the authors suggested a mechanism that eastward neutral winds in the F-region ionosphere across the dawn terminator cause the enhancement of zonal electric field. This is similar to the generation mechanism for the prereversal enhancement of the zonal electric field during sunset period. However, this mechanism may explain the enhancement in Sq current around the dawn terminator, but it cannot cause the enhancement in Pc 3 amplitude. The authors need to explain how the neutral winds can give rise to the electric field oscillations at the Pc 3 frequency. I guess that this mechanism is difficult to explain the equatorial enhancement of Pc 3 amplitude, therefore, the authors should suggest other possible mechanism.

Line 11-15 of page 14: The authors described that the Earth-ionosphere waveguide propagation model is inconsistent with the phase lag of Pc 5 pulsations at the dip equator. However, according to Shinohara et al. [1998], the phase lag can be explained by the induction effect of the equatorial enhanced ionospheric current above the good conductor Earth. The source of the equatorial ionospheric current can be the electric field propagating from high latitude by the Earth-ionosphere waveguide mode. On the other hand, it is not clear whether or not the MHD wave propagation model can cause the phase lag at the dip equator.

Technical corrections:

Line 3 of page 2: internally to the magnetosphere –> internally in the magnetosphere

Line 4 of page 6: Please describe the period range of bandpass filter for Pc 3 and Pc 5 pulsations in the text.

Line 8-9 of page 8: Please state the duration of the Pc 3 and Pc 5 events. Is the duration fixed? For example, if Pc 3 pulsations continued during 30 minutes, are they counted as one event?

Subsection 3.2: Was the power spectral analysis applied to bandpass-filtered data? I recommend that the authors explain the analysis procedure made by Roy and Rao [1998] briefly.

Figure 5, 6, 7: In this manuscript, the amplification factor is defined as the amplitude ratio of pulsations at CUI to those at other stations, so greater values than one correspond to equatorial depression and smaller values than one correspond to equatorial amplification. However, this definition is a little bit confusing. In addition, different definition is used in Figure 8. Thus, I suggest that the authors use the amplitude ratio of (other stations)/CUI all through the manuscript.

Figure 10: Please plot the H/D amplitude ratio or the PRM(D)/CUI(H) ratio in this figure.

---

## Referee Comment (RC2) · Anonymous Referee #2 · 9 Sep 2019

This manuscript statistically analyzes the latitudinal dependence of the wave characteristics of Pc3 and Pc5 pulsations using magnetic variation data from multiple equatorial stations. The authors newly indicate that attenuation in the Pc3 amplitude around noon, especially, at the dip equator and for wave periods shorter than $\sim$ 35 S, and that Pc5 amplitude also shows a slight depression at the dip equator while Pc5 amplitude basically shows an enhancement at all equatorial stations. In addition, the authors indicate that the Pc3 amplitude shows an amplification at dawn sector in H component rather than D component.

These results of the analysis include some significant new findings. However, some explanations for these observational facts need more improvement. Thus, I suggest that this manuscript is fairly important, which can be published in Annales of Geophysicae after revision reflected following specific comments.

(1) Page 13, Line 3,

The authors explained the amplitude depression of Pc5 at equatorial region by the propagation model proposed by Chi et al. (2001). However, regarding this paper, there were some discussions between Chi et al. and Kikuchi and Araki, while the author's interpretation seems to be valid explanation. It would be better to refer the following papers to consider the adequacy of the explanations.

(a) Kikuchi, T., and T. Araki, Comment on "Propagation of the preliminary reverse impulse of sudden commencements to low latitudes" by P. J. Chi et al.,J. Geophys. Res.,107(A12), 1473, doi:10.1029/2001JA009220, 2002.

(b) Chi, P. J., et al., Reply to comment by T. Kikuchi and T. Araki on "Propagation of the preliminary reverse impulse of sudden commencements to low latitudes," J. Geophys. Res.,107(A12), 1474, doi:10.1029/2002JA009369, 2002.

(2) Page 18, Line 32.

The authors stated Pc5 amplitude depression and phase lag at the dip equator are not predicted by the horizontal transmission. However, these features well correspond with that of Pi2 pulsation which was indicated by Shinohara et al., (1997, and 1998). They explained that the transmitted electric fields from the polar ionosphere could cause the dayside Pi2 with amplitude depression and phase lag due to the high ionospheric conductivity at dip equator. It would be better to take account these previous studies in considering the interpretation of the observational facts of Pc5 pulsation.

(3) Page 18, Line 7; Figure 11 and Page 19, Line 5,

The authors suggest that the amplitude enhancement in H component of Pc3 at dawn terminator could be explained by the secondary electric field shown in Figure 11. However, in this model, the secondary electric field is generated by the neutral thermospheric wind. In this scenario, the secondary electric field should be modulated by the temporal variation of the neutral wind, not by the electric field associated with the Pc3. It seems to be less convinced. The authors need to explain how the secondary electric field at the dawn terminator affect the amplitude enhancement of Pc3 which is imposed on the dawn ionosphere.

Minor comments:

Figure3, 4, 5(a), 9

It would be better to add local time on horizontal axis.

---

## Author Comment (AC1) · 9 Oct 2019

The authors gratefully thank to the Referee for the constructive comments and recommendations which definitely help to improve the readability and quality of the manuscript. Following the editors' suggestion, the response is structured as follows: comments from the referee, authors' response, and changes to be made to the manuscript.

Specific comments:

Section 5: To explain the equatorial enhancement of Pc 3 pulsations in the H compo-
nent at the dawn terminator, the authors suggested a mechanism that eastward neutral winds in the F-region ionosphere across the dawn terminator cause the enhancement of zonal electric field. This is similar to the generation mechanism for the prereversal enhancement of the zonal electric field during the sunset period. However, this mechanism may explain the enhancement in Sq current around the dawn terminator, but it cannot cause the enhancement in Pc3 amplitude. The authors need to explain how the neutral winds can give rise to the electric field oscillations at the Pc 3 frequency. I guess that this mechanism is difficult to explain the equatorial enhancement of Pc3 amplitude, therefore, the authors should suggest other possible mechanism.

Reply: Analyzing the scientific literature on how ULF wave fields couple to the ionosphere to generate secondary fields modulated by background conductivity during sunrise hours, we are now proposing that this H-component enhancement effect in Pc3 pulsations at the dawn terminator should be mainly related to the mode of propagation and incidence of these waves on the ionosphere. For example, the model of fast mode incidence on the ionosphere described by Alperovich and Fedorov [2007, p. 284-297] predicts that compressional waves in the Pc3 frequency band are those expected to produce the strongest structure of zonal electric fields and currents in the vicinity of the dip equator (mainly during the equinoxes) because its effect increases with increasing frequency. In addition, this model also allows external electric fields other than those carried by the waves (eg, induced by dynamo action) to contribute substantially to increased zonal currents and consequent signal amplification on the ground. We understand that the dynamo mechanism suggested in our text, in which the F region thermo-atmospheric winds drive zonal electric currents, can be applied to explain the Pc3s enhancement near sunrise, given the large longitudinal variations in electrodynamic processes in the South American equatorial ionosphere. However, we still need to assume that our Pc3 events occurring around sunrise are also related to compressional MHD waves vertically incident on the ionosphere, as recognized for the Pc3 events around noon.

Changes to be made to the manuscript: This discussion will be included in the manuscript, including the summary and conclusions. In addition, a new reference will be added to the text: Alperovich, L.S., Fedorov, E.N.: Hydromagnetic Waves in the Magnetosphere and the Ionosphere. Series: Astrophysics and Space Science Library, Vol. 353, 2009, XXIV, 418 pp., doi:10.1007/978-1-4020-6637-5, 2007.

Line 11-15 of page 14: The authors described that the Earth-ionosphere waveguide propagation model is inconsistent with the phase lag of Pc 5 pulsations at the dip equator. However, according to Shinohara et al. [1998], the phase lag can be explained by the induction effect of the equatorial enhanced ionospheric current above the good conductor Earth. The source of the equatorial ionospheric current can be the electric field propagating from high latitude by the Earth-ionosphere waveguide mode. On the other hand, it is not clear whether or not the MHD wave propagation model can cause the phase lag at the dip equator.

Reply: The reviewer is correct in stating that incident waves coming horizontally from the poles through the Earth-ionosphere waveguide can cause phase delays in magnetic signals at the dip equator relative to off-equatorial regions. This information will be added to our manuscript, as these phase delays can be explained by invoking the induction effect with the model devised by Shinohara et al. [1998]. On the other hand, a discussion on the possibility that the MHD wave propagation model could also cause phase delay in the dip equator had already been presented in our text (lines 9-12 of page 13). We had explained that the screening effect induced by enhanced ionospheric conductivity in the equator is also recognized for causing amplitude depression and phase delays on MHD signals incident from the magnetosphere.

Changes to be made to the manuscript: Corrections will be made to the manuscript to include the induction effect discussed by Shinohara et al. [1998].

Technical corrections:

Line 3 of page 2: internally to the magnetosphere -> internally in the magnetosphere

Reply/Changes in manuscript: It will be corrected in the text.

Line 4 of page 6: Please describe the period range of bandpass filter for Pc 3 and Pc 5 pulsations in the text.

Reply/Changes in manuscript: Information will be included in the text.

Line 8-9 of page 8: Please state the duration of the Pc 3 and Pc 5 events. Is the duration fixed? For example, if Pc 3 pulsations continued during 30 minutes, are they counted as one event?

Reply/Changes in manuscript: Most Pc3 events analyzed have duration of about 3 minutes, while Pc5s presented durations shorter than 24 minutes. For all cases, we considered them as a single event, regardless of their duration. This information will be included in the text.

Subsection 3.2: Was the power spectral analysis applied to bandpass-filtered data? I recommend that the authors explain the analysis procedure made by Roy and Rao [1998] briefly.

Reply/Changes in manuscript: Yes, the power spectral analysis was applied to bandpass-filtered data. This information will be included in the text and we will also give more details about the procedure performed by Roy and Rao [1998].

Figure 5, 6, 7: In this manuscript, the amplification factor is defined as the amplitude ratio of pulsations at CUI to those at other stations, so greater values than one correspond to equatorial depression and smaller values than one correspond to equatorial amplification. However, this definition is a little bit confusing. In addition, different definition is used in Figure 8. Thus, I suggest that the authors use the amplitude ratio of (other stations)/CUI all through the manuscript.

Reply/Changes in manuscript: We accept the referee's suggestion and therefore the definition of the amplification factor (and Figures 6-7) will be changed throughout the manuscript.

Figure 10: Please plot the H/D amplitude ratio or the PRM(D)/CUI(H) ratio in this figure.

Reply/Changes in manuscript: The PRM(D)/CUI(H) ratio was obtained for each Pc3 event and will be plotted in Figure 10 along with the PRM(H)/CUI(H) ratio. Text will be included to discuss the results.Âă
* * *

---

## Author Comment (AC2) · 9 Oct 2019

We thank the referee for the careful and insightful review of our manuscript. Following the editors 'suggestion, the response is structured as follows: comments from the referee, authors' response, and changes to be made to the manuscript.

(1) Page 13, Line 3,

The authors explained the amplitude depression of Pc5 at equatorial region by the propagation model proposed by Chi et al. (2001). However, regarding this paper, there were some discussions between Chi et al. and Kikuchi and Araki, while the author's

interpretation seems to be valid explanation. It would be better to refer the following papers to consider the adequacy of the explanations.

(a) Kikuchi, T., and T. Araki, Comment on "Propagation of the preliminary reverse impulse of sudden commencements to low latitudes" by P. J. Chi et al.,J. Geophys. Res.,107(A12), 1473, doi:10.1029/2001JA009220, 2002.

(b) Chi, P. J., et al., Reply to comment by T. Kikuchi and T. Araki on "Propagation of the preliminary reverse impulse of sudden commencements to low latitudes," J. Geophys. Res.,107(A12), 1474, doi:10.1029/2002JA009369, 2002.

Reply/Changes in manuscript: We will add to the text the references suggested by the reviewer. Note that the discussion between Chi et al. and Kikuchi and Araki on the origin of equatorial PRIs support our conclusions on the Pc5s features observed in the Brazilian equatorial region.

(2) Page 18, Line 32.

The authors stated Pc5 amplitude depression and phase lag at the dip equator are not predicted by the horizontal transmission. However, these features well correspond with that of Pi2 pulsation which was indicated by Shinohara et al., (1997, and 1998). They explained that the transmitted electric fields from the polar ionosphere could cause the dayside Pi2 with amplitude depression and phase lag due to the high ionospheric conductivity at dip equator. It would be better to take account these previous studies in considering the interpretation of the observational facts of Pc5 pulsation.

Reply: The reviewer is correct in stating that incident waves coming horizontally from the poles through the Earth-ionosphere waveguide can cause phase delays in magnetic signals at the dip equator relative to off-equatorial regions. This information will be added to our manuscript, as these phase delays can be explained by invoking the induction effect with the model devised by Shinohara et al. [1997, 1998]. However, it is not clear how this model could account for the Pc5 amplitude depression observed in

our data during daytime at the dip equator. This is because the amplitude depression reported by Shinohara et al. (1997) occurred only for nighttime Pc5 pulsations. In this case, the authors resort to the MHD compressive propagation model (page 2282 of that paper), where the waves are directly incident on the equatorial ionosphere and have their amplitude damped due to screening effects. Thus, we maintain our proposal that the two propagation models should be operating interactively in our region of study.

Changes to be made to the manuscript: Corrections will be made to the manuscript to include the induction effect discussed by Shinohara et al. [1998].

(3) Page 18, Line 7; Figure 11 and Page 19, Line 5,

The authors suggest that the amplitude enhancement in H component of Pc3 at dawn terminator could be explained by the secondary electric field shown in Figure 11. However, in this model, the secondary electric field is generated by the neutral thermospheric wind. In this scenario, the secondary electric field should be modulated by the temporal variation of the neutral wind, not by the electric field associated with the Pc3. It seems to be less convinced. The authors need to explain how the secondary electric field at the dawn terminator affect the amplitude enhancement of Pc3 which is imposed on the dawn ionosphere.

Reply: Analyzing the scientific literature on how ULF wave fields couple to the ionosphere to generate secondary fields modulated by background conductivity during sunrise hours, we are now proposing that this H-component enhancement effect in Pc3 pulsations at the dawn terminator should be mainly related to the mode of propagation and incidence of these waves on the ionosphere. For example, the model of fast mode incidence on the ionosphere described by Alperovich and Fedorov [2007, p. 284-297] predicts that compressional waves in the Pc3 frequency band are those expected to produce the strongest structure of zonal electric fields and currents in the vicinity of the dip equator (mainly during the equinoxes) because its effect increases with increasing frequency. In addition, this model also allows external electric fields other than

those carried by the waves (eg, induced by dynamo action) to contribute substantially to increased zonal currents and consequent signal amplification on the ground. We understand that the dynamo mechanism suggested in our text, in which the F region thermo-atmospheric winds drive zonal electric currents, can be applied to explain the Pc3s enhancement near sunrise, given the large longitudinal variations in electrodynamic processes in the South American equatorial ionosphere. However, we still need to assume that our Pc3 events occurring around sunrise are also related to compressional MHD waves vertically incident on the ionosphere, as recognized for the Pc3 events around noon.

Changes to be made to the manuscript: This discussion will be included in the manuscript, including the summary and conclusions. In addition, a new reference will be added to the text: Alperovich, L.S., Fedorov, E.N.: Hydromagnetic Waves in the Magnetosphere and the Ionosphere. Series: Astrophysics and Space Science Library, Vol. 353, 2009, XXIV, 418 pp., doi:10.1007/978-1-4020-6637-5, 2007.

Minor comments:

Figure3, 4, 5(a), 9

It would be better to add local time on horizontal axis.

Reply/Changes in the manuscript: Local time will be added on the horizontal axes of these figures.
* * *

---

## Author Response (AR1)

[revised manuscript text omitted]

**Point-by-point reply to the comments of Reviewer#1**

The authors gratefully thank to the Referee for the constructive comments and recommendations which definitely help to improve the readability and quality of the manuscript. Following the editors' suggestion, the response is structured as follows: comments from the referee, authors' response, and changes made to the manuscript.

==Changes related to this Referee's comments are colored blue and brown in the marked version of the manuscript. Blue was used for changes requested only by this referee and brown for changes suggested by both referees. All these changes in the text are also highlighted in yellow.==

Specific comments:

Section 5: To explain the equatorial enhancement of Pc 3 pulsations in the H component at the dawn terminator, the authors suggested a mechanism that eastward neutral winds in the F-region ionosphere across the dawn terminator cause the enhancement of zonal electric field. This is similar to the generation mechanism for the prereversal enhancement of the zonal electric field during sunset period. However, this mechanism may explain the enhancement in Sq current around the dawn terminator, but it cannot cause the enhancement in Pc 3 amplitude. The authors need to explain how the neutral winds can give rise to the electric field oscillations at the Pc 3 frequency. I guess that this mechanism is difficult to explain the equatorial enhancement of Pc 3 amplitude, therefore, the authors should suggest other possible mechanism.

Reply: Analyzing the scientific literature on how ULF wave fields couple to the ionosphere to generate secondary fields modulated by background conductivity during sunrise hours, we are proposing that this H-component enhancement effect in Pc3 pulsations at the dawn terminator should be mainly related to the mode of propagation and incidence of these waves on the ionosphere. For example, the model of fast mode incidence on the ionosphere described by Alperovich and Fedorov [2007, p. 284-297] predicts that compressional waves in the Pc3 frequency band are those expected to produce the strongest structure of zonal electric fields and currents in the vicinity of the dip equator (mainly during the equinoxes) because its effect increases with increasing frequency. In addition, this model also allows external electric fields other than those carried by the waves (e.g., induced by dynamo action) to contribute substantially to increased zonal currents and consequent signal amplification on the ground. We understand that the dynamo mechanism suggested in our text, in which the F region thermo-atmospheric winds drive zonal electric currents, can be applied to explain the Pc3s' enhancement near sunrise, given the large longitudinal variations in electrodynamic processes in the South American equatorial ionosphere. However, we still need to assume that our Pc3 events occurring around sunrise are also related to compressional MHD waves vertically incident on the ionosphere, as recognized for the Pc3 events around noon.

Changes in manuscript: This discussion has been included in the manuscript on page 20 (lines 9-21) and partly in the summary (lines 23-24 on page 1) and conclusions (lines 28-29 on page 21). In addition, a new reference has been added (lines 18-19 on page 22).

5   Line 11-15 of page 14: The authors described that the Earth-ionosphere waveguide propagation model is inconsistent with the phase lag of Pc 5 pulsations at the dip equator. However, according to Shinohara et al. [1998], the phase lag can be explained by the induction effect of the equatorial enhanced ionospheric current above the good conductor Earth. The source of the equatorial ionospheric current can be the electric field propagating from high latitude by the Earth-ionosphere waveguide mode. On the other hand, it is not clear
10   whether or not the MHD wave propagation model can cause the phase lag at the dip equator.

Reply: The reviewer is correct in stating that incident waves coming horizontally from the poles through the Earth-ionosphere waveguide can cause phase delays in magnetic signals at the dip equator relative to off-equatorial regions. This information was added to our manuscript, as these phase delays can be explained by invoking the induction effect with the model devised by Shinohara et al. [1998]. On the other hand, a
15   discussion on the possibility that the MHD wave propagation model could also cause phase delay in the dip equator had already been presented in our text (lines 25-27 of page 14). We had explained that the screening effect induced by enhanced ionospheric conductivity in the equator is also recognized for causing amplitude depression and phase delays on MHD signals incident from the magnetosphere.

Changes in manuscript: Corrections were made on page 15 (lines 18-21) of the manuscript to include the
20   induction effect discussed by Shinohara et al. [1998]. Other changes also related to this point were also made elsewhere in the text (page 16, line 3; page 21, lines 18-19).

Technical corrections:

Line 3 of page 2: internally to the magnetosphere -> internally in the magnetosphere

25   Reply/Changes in manuscript: Fixed in line 3 of page 2.

Line 4 of page 6: Please describe the period range of bandpass filter for Pc 3 and Pc 5 pulsations in the text.

Reply/Changes in manuscript: Information included on page 7 (lines 6-7).

Line 8-9 of page 8: Please state the duration of the Pc 3 and Pc 5 events. Is the duration fixed? For example, if Pc 3 pulsations continued during 30 minutes, are they counted as one event?

Reply/Changes in manuscript: Most Pc3 events analyzed have duration of about 3 minutes, while Pc5s presented durations shorter than 24 minutes. For all cases, we considered them as a single event, regardless of their duration. This information was included on page 9 (lines 13-14).

Subsection 3.2: Was the power spectral analysis applied to bandpass-filtered data? I recommend that the authors explain the analysis procedure made by Roy and Rao [1998] briefly.

Reply/Changes in manuscript: Yes, the power spectral analysis was applied to bandpass-filtered data. This information was included on pages 9 (lines 17-18) and 10 (lines 1-2 and 6-7), where we also give more details about the procedure performed by Roy and Rao [1998].

Figure 5, 6, 7: In this manuscript, the amplification factor is defined as the amplitude ratio of pulsations at CUI to those at other stations, so greater values than one correspond to equatorial depression and smaller values than one correspond to equatorial amplification. However, this definition is a little bit confusing. In addition, different definition is used in Figure 8. Thus, I suggest that the authors use the amplitude ratio of (other stations)/CUI all through the manuscript.

Reply/Changes in manuscript: We accepted the referee's suggestion and therefore the definition of the amplification factor was changed throughout the manuscript (page 10, lines 15-16) and also in the modified Figures 6 and 7.

Figure 10: Please plot the H/D amplitude ratio or the PRM(D)/CUI(H) ratio in this figure.

Reply/Changes in manuscript: The PRM(D)/CUI(H) ratio was obtained for each Pc3 event and was plotted in the modified Figure 10 along with the PRM(H)/CUI(H) ratio. Text was included to present and discuss the results on pages 17 (lines 8-13 and 15-16) and 18 (3-4).

**Point-by-point reply to the comments of Reviewer#2**

We thank the referee for the careful and insightful review of our manuscript. Following the editors 'suggestion, the response is structured as follows: comments from the referee, authors' response, and
5   changes made to the manuscript.

Changes related to this Referee's comments are colored red and brown in the marked version of the manuscript. Red was used for changes requested only by this referee and brown for changes suggested by both referees. All these changes in the text are also highlighted in yellow.

10   (1) Page 13, Line 3,

The authors explained the amplitude depression of Pc5 at equatorial region by the propagation model proposed by Chi et al. (2001). However, regarding this paper, there were some discussions between Chi et al. and Kikuchi and Araki, while the author's interpretation seems to be valid explanation. It would be better to refer the following papers to consider the adequacy of the explanations.

15   (a) Kikuchi, T., and T. Araki, Comment on "Propagation of the preliminary reverse impulse of sudden commencements to low latitudes" by P. J. Chi et al.,J. Geophys. Res.,107(A12), 1473, doi:10.1029/2001JA009220, 2002.

(b) Chi, P. J., et al., Reply to comment by T. Kikuchi and T. Araki on "Propagation of the preliminary reverse impulse of sudden commencements to low latitudes," J. Geophys. Res.,107(A12), 1474,
20   doi:10.1029/2002JA009369, 2002.

Reply/Changes in manuscript: We have added to the text the discussion and references suggested by the reviewer (page 14, lines 20-21; page 22, lines 23-25; page 24, lines 1-2). Note that the discussion between Chi et al. and Kikuchi and Araki on the origin of equatorial PRIs support our conclusions on the Pc5s features observed in the Brazilian equatorial region.

(2) Page 18, Line 32.

The authors stated Pc5 amplitude depression and phase lag at the dip equator are not predicted by the horizontal transmission. However, these features well correspond with that of Pi2 pulsation which was indicated by Shinohara et al., (1997, and 1998). They explained that the transmitted electric fields from the polar ionosphere could cause the dayside Pi2 with amplitude depression and phase lag due to the high ionospheric conductivity at dip equator. It would be better to take account these previous studies in considering the interpretation of the observational facts of Pc5 pulsation.

Reply: The reviewer is correct in stating that incident waves coming horizontally from the poles through the Earth-ionosphere waveguide can cause phase delays in magnetic signals at the dip equator relative to off-equatorial regions. This information was added to our manuscript, as these phase delays can be explained by invoking the induction effect with the model devised by Shinohara et al. [1997, 1998]. However, it is not clear how this model could account for the Pc5 amplitude depression observed in our data during daytime at the dip equator. This is because the amplitude depression reported by Shinohara et al. (1997) occurred only for nighttime Pc5 pulsations. In this case, the authors resort to the MHD compressive propagation model (page 2282 of that paper), where the waves are directly incident on the equatorial ionosphere and have their amplitude damped due to screening effects. Thus, we maintain our proposal that the two propagation models should be operating interactively in our region of study.

Changes in manuscript: Corrections were made on page 15 (lines 18-21) of the manuscript to include the induction effect discussed by Shinohara et al. [1998]. Other changes also related to this point were also made elsewhere in the text (page 16, line 3; page 21, lines 18-19).

(3) Page 18, Line 7; Figure 11 and Page 19, Line 5,

The authors suggest that the amplitude enhancement in H component of Pc3 at dawn terminator could be explained by the secondary electric field shown in Figure 11. However, in this model, the secondary electric field is generated by the neutral thermospheric wind. In this scenario, the secondary electric field should be modulated by the temporal variation of the neutral wind, not by the electric field associated with the Pc3. It seems to be less convinced. The authors need to explain how the secondary electric field at the dawn terminator affect the amplitude enhancement of Pc3 which is imposed on the dawn ionosphere.

Reply: Analyzing the scientific literature on how ULF wave fields couple to the ionosphere to generate secondary fields modulated by background conductivity during sunrise hours, we are now proposing that this

H-component enhancement effect in Pc3 pulsations at the dawn terminator should be mainly related to the mode of propagation and incidence of these waves on the ionosphere. For example, the model of fast mode incidence on the ionosphere described by Alperovich and Fedorov [2007, p. 284-297] predicts that compressional waves in the Pc3 frequency band are those expected to produce the strongest structure of zonal electric fields and currents in the vicinity of the dip equator (mainly during the equinoxes) because its effect increases with increasing frequency. In addition, this model also allows external electric fields other than those carried by the waves (eg, induced by dynamo action) to contribute substantially to increased zonal currents and consequent signal amplification on the ground. We understand that the dynamo mechanism suggested in our text, in which the F region thermo-atmospheric winds drive zonal electric currents, can be applied to explain the Pc3s enhancement near sunrise, given the large longitudinal variations in electrodynamic processes in the South American equatorial ionosphere. However, we still need to assume that our Pc3 events occurring around sunrise are also related to compressional MHD waves vertically incident on the ionosphere, as recognized for the Pc3 events around noon.

Changes in manuscript: This discussion has been included in the manuscript on page 20 (lines 9-21) and partly in the summary (lines 23-24 on page 1) and conclusions (lines 28-29 on page 21). In addition, a new reference has been added (lines 18-19 on page 22).

Minor comments:

Figure3, 4, 5(a), 9

It would be better to add local time on horizontal axis.

Reply/Changes in the manuscript: Local time was added on the horizontal axes of the modified version of these figures.